# Evidence for Dark Energy Driven by Star Formation: Information Dark Energy?

**DOI:** 10.3390/e27020110

**Published:** 2025-01-23

**Authors:** Michael Paul Gough

**Affiliations:** School of Engineering and Informatics, University of Sussex, Brighton BN1 9QT, UK; m.p.gough@sussex.ac.uk

**Keywords:** Landauer’s principle, dark energy, dark energy experiments

## Abstract

Evidence is presented for dark energy resulting directly from star formation. A survey of stellar mass density measurements, SMD(*a*), as a function of universe scale size *a*, was found to be described by a simple CPL w_0_ − w_a_ parameterisation that was in good agreement with the dark energy results of *Planck* 2018, Pantheon+ 2022, the Dark Energy Survey 2024, and the Dark Energy Spectroscopic Instrument 2024. The best-fit CPL values found were *w*_0_ = −0.90 and *w*_a_ = −1.49 for SMD(*a*), and *w*_0_ = −0.94 and *w*_a_ = −0.76 for SMD(*a*)^0.5^, corresponding with, respectively, good and very good agreement with all dark energy results. The preference for SMD(*a*)^0.5^ suggests that it is the temperature of astrophysical objects that determines the dark energy density. The equivalent energy of the information/entropy of gas and plasma heated by star and structure formations is proportional to temperature, and is then a possible candidate for such a dark energy source. Information dark energy is also capable of resolving many of the problems and tensions of ΛCDM, including the cosmological constant problem, the cosmological coincidence problem, and the *H*_0_ and σ_8_ tensions, and may account for some effects previously attributed to dark matter.

## 1. Introduction

The ΛCDM model has been very successful despite our inability to account for either the cosmological constant, Λ, or cold dark matter, CDM. It is well known that the natural value of Λ is a factor of ~10^120^ different from the observed value. Also, there has not been a single confirmed detection of a CDM particle of any type, including WIMP, axion, etc. In addition, when using ΛCDM to extrapolate from early universe measurements to the late universe, there appears to be a significant difference, or tension, with the Hubble constant, *H_o_*, and with the *σ_8_* matter fluctuation parameter measured today. Therefore, despite the success of ΛCDM, we are encouraged to also consider alternative explanations. Here, we consider the role of information energy.

Information must play a significant physical role in the universe. Rolf Landauer [1,2] showed that “*Information is Physical*”, as each bit of information in a system at temperature *T* has an energy equivalence of kB *T* ln(2). Laboratory experiments have proven the Landauer information energy equivalence [3,4,5,6]. John Wheeler [7] even considered information to be more fundamental than matter, with all things physical being information-theoretic in origin, a view encapsulated by his famous slogan “*It from Bit*”. In the same vein, Anton Zeilinger [8] proposed a “*Foundational Principle*” whereby the attributes of all particles at their most fundamental level correspond with elemental systems, each with just one classical bit or quantum qubit of information.

A strong similarity was found [9] between information energy and a cosmological constant. Consider a Zeilinger elemental bit of a particle attribute in a simple universe, without star formation. The Landauer equivalent energy of such a bit has been shown to be defined exactly the same as, and have the same value as, the characteristic energy of a cosmological constant [9,10].

These ideas of information have encouraged research [11,12,13,14,15] into the possible role information energy may play as a source of dark energy. Such a source would be governed by the product of the source bit number total, *N* [16,17], with a typical source temperature, *T.* Previously, the time history of alternative dark energy contributions was compared with the generally assumed cosmological constant Λ. By definition, Λ has a constant energy density, or a total energy proportional to *a*^3^, where *a* is the scale size of the universe, given by *a* =1/(1 + *z*); *z* is redshift; and *a* = 1 today. A time history of information energy was obtained by combining the stellar mass density history, SMD(*a*), for *T*, with the holographic principle [18,19,20] for *N*. During late-universe times, *z* < 1.35; the *NT* product was found to also vary as *a*^3^ with a near-constant information energy density, emulating a cosmological constant. During earlier times, *z* > 1.35; the steeper gradient would provide a means by which the information dark energy could be differentiated from a cosmological constant and effectively falsified [15].

However, the universe information content is well below the holographic bound (~10^124^ bits) and, so far, the holographic principle has only been verified to apply to black holes at that bound. The approach taken in the present work was to show that information energy could account for dark energy history based solely on the measured SMD(*a*), without invoking the holographic principle. Compared with the most recent work [15], the approach here is simpler and more natural. Moreover, the predicted time history of this information dark energy is provided in the same form as the results from dark energy measurements, enabling a direct comparison between theory and experiments.

## 2. Information Dark Energy (IDE)

The equation of state parameter, *w*, of dark energy sets the time variation in the dark energy density as being proportional to *a*^−3(1+*w*)^. While baryon and dark matter energy densities vary as *a*^−3^ and *w* = 0, the energy density of a cosmological constant is, by definition, constant, unchanging as *a*^0^ and *w* = −1. In contrast, a dynamic form of dark energy varies at different rates at different times. In order to take any such time variation into account, most dark energy studies have adopted the CPL [21] parameters *w*_0_ and *w*_a_ for a variable equation of state parameter, *w*(*a*) = *w*_0_ + (1 − *a*) *w*_a_. This provides *w*(*a*) with a smooth variation from the very early value of *w*_0_ + *w*_a_ to the present value of *w*_0_. While there is no reason to expect that any source of dark energy with a time varying *w*(*a*) can be fully described by CPL parameters, it does have the advantage of being simple and, for that reason, it is widely applied to studies of dark energy. CPL provides a common testing ground between experiments and theory. Dark energy measurements were originally expected to strengthen the cosmological constant hypothesis by converging on the values of *w*_0_ = −1 and *w*_a_ = 0, but recent dark energy measurements [22,23,24] clearly tend towards a dynamic dark energy description, with *w*_0_ > −1 and *w*_a_ < 0.

As the main source of information dark energy, IDE(*a*) is the information energy of hot gases and plasma heated by star and general structure formations [15], we must consider the stellar mass density, SMD(*a*), as a function of universe scale size, *a*. Figure 1 provides a survey of SMD(*a*) measurements in units of solar masses per cubic co-moving megaparsec. A total of 121 SMD(*a*) measurements from 27 published sources [25,26,27,28,29,30,31,32,33,34,35,36,37,38,39,40,41,42,43,44,45,46,47,48,49,50,51] (some of which were covered in a review [52]) are plotted in Figure 1 on a logarithmic scale of stellar mass density against the logarithm of universe scale size, *a*. Note that only values of *a* > 0.2 are used in the analysis below. This is because values of a < 0.2 would correspond to times when the information energy density was so much weaker, <<1%, of the matter energy density (varying as *a*^−3^), and thus could not affect universe expansion measurements. Dark energy is only evident from measurements of the expansion rate history.

As SMD(*a*) is a universe-wide average mass density and IDE(*a*) is a universe-wide average information energy density—both effectively energy densities—we expect IDE(*a*) to vary at some power, *p*, of proportionality as SMD(*a*)*^p^*. In this way, we can account for time variations in the *NT* product (without recourse to employing the holographic principle, as it was previously).

The Friedmann equation [53] describes the Hubble parameter *H*(*a*) in terms of the Hubble constant, *H*_0_, and dimensionless density parameters, Ω, expressed as a fraction of today’s total energy density. We can assume that the curvature term is zero and that the radiation term has been negligible for some time. The ΛCDM model is then given by Equation (1). The equivalent IDE model is then described by Equation (2), where the present fractional energy density contributions are Ω_tot_ from all matter (baryons + dark matter); Ω_Λ_ is the cosmological constant and Ω_IDE_ is information dark energy (IDE).**ΛCDM:**      (*H*(*a*)/*H_0_*)^2^ = Ω_tot_ *a*^−3^ + Ω_Λ_(1)**IDE:**      (*H*(*a*)/*H_0_*)^2^ = Ω_tot_ *a*^−3^ + Ω_IDE_ (SMD(*a*)/SMD(1.0))^p^(2)

Note that the energy density terms Ω_tot_ and Ω_IDE_ are energy density fractions, assuming the mc^2^ energy equivalence of mass, and the Landauer, kB *T* ln(2), energy equivalence of information. No matter needs to be destroyed, nor information erased, nor such processes identified in order to use these energy equivalences in these equations.

It seems most natural to assume that the heating of gases and plasmas is directly proportional to the amount of star formation, *p* = +1. However, we see in Section 4 that there are many general cases in the universe where temperatures closely follow the square root of mass, corresponding with *p* = +0.5. As the Landauer information equivalent energy is proportional to temperature, we therefore examined both cases, IDE(*a*) α SMD(*a*) and IDE(*a*) α SMD(*a*)^0.5^, as shown in Figure 2.

The SMD and SMD^0.5^ data illustrated in Figure 2 were tested against all CPL *w*_0_ and *w*_a_ combinations; *w*_0_ values in the range of +0.9 to −3.0 in steps of 0.01 were tested for each *w*_a_ value in the range of +0.9 to −3.5, also in steps of 0.01. As this was a log–log plot, any curve corresponding with a given *w*_0_ and *w*_a_ combination was fixed in the log(*a*) abscissa direction, but could be moved in the ordinate direction to find the best fit. The best fits for each *w*_0_ and *w*_a_ combination were determined by the position in the ordinate direction that gave the minimum value of the residual sum of squares (RSS). Then, that *w*_0_ and *w*_a_ combination was assigned an R^2^ coefficient of determination, given by R^2^ = 1 − (RSS/TSS), where TSS is the total sum of squares.

The *w*_0_ and *w*_a_ combinations that provided the highest R^2^ (or minimum RSS), both with maximum R^2^ values of 0.93, are shown in Figure 2 for the two cases of SMD and SMD^0.5^.

In Figure 3, each of the 121 SMD(*a*) measurements are plotted against the SMD(*a*) values predicted by the best-fit CPL curve at the measured *a* value. In both cases, the measured and predicted sets of values were related, with a Pearson correlation coefficient value of *r* = 0.92.

The maximum R^2^ coefficients are plotted in Figure 4 on a colour scale for all of the above *w*_0_ − *w*_a_ tested combinations to show the extent of fit for different *w*_0_ and *w*_a_ values for both cases.

## 3. Comparison of IDE Prediction with Experimental Measurements

In this section, we compare the IDE(*a*) contribution with several experimental measurements of dark energy. In Figure 5 and Figure 6 we illustrate the expected information dark energy contribution in *w*_0_ − *w*_a_ space for the two cases of IDE(*a*) α SMD(*a*) and IDE(*a*) α SMD(*a*)^0.5^ by re-plotting the R^2^ > 0.86 and R^2^ > 0.92 contours from Figure 4 in two shades of red, adjusted to the scales of the various published *w*_0_ − *w*_a_ experimental plots. Both IDE(a) predictions are compared in these figures directly with the results of *Planck* 2018 [54], Pantheon+ 2022 [22], the Dark Energy Survey 2024 [23], and the Dark Energy Spectroscopic Instrument (DESI) 2024 [24].

These combined plots allowed the region of predicted IDE(a) in *w*_0_ − *w*_a_ space to be directly compared with the *w*_0_ − *w*_a_ space required to explain the experimental measurements of dark energy. In each measurement plot, there were several combinations of experimental techniques and different colours were used to differentiate between the different combinations, each with two shades per combination, corresponding with the 68% (stronger colour) and 95% (lighter colour) likelihood contours. The combinations included several different techniques, such as the cosmic microwave background, baryon acoustic oscillations, weak lensing, etc. The reader is referred to the cited publications for more information on these techniques and how data from the techniques were combined.

The IDE predicted *w*_0_ − *w*_a_ spaces for IDE(*a*) α SMD(*a*) in Figure 5 and Figure 6 clearly show a strong overlap with the *Planck* plots of likelihood space, and lie close to, but with less direct overlap with, the more recent measurements of Pantheon+, the Dark Energy Survey, and the Dark Energy Spectrographic Instrument. However, the predicted IDE *w*_0_ − *w*_a_ spaces for IDE(*a*) α SMD(*a*)^0.5^ show an even stronger overlap in all plots. The best-fit IDE, R^2^ > 0.92, overlaps the deeper-colour 68% likelihood area of all experimental data combinations in all plots.

## 4. Discussion

### 4.1. IDE Can Account for the Observed Dark Energy

While we might well expect a direct proportionality between IDE(*a*) and SMD(*a*), to exhibit the observed reasonable agreement in *w*_0_ − *w*_a_ space, we find that SMD(*a*)^0.5^ provides an even better, fuller, agreement. In this case, two decades of SMD(a) mass density provide only one decade of IDE(a). Information energy, *N* kB *T* ln(2), is proportional to temperature and we note that there are many cases in the universe where the temperature of objects scale approximately with the (object mass)^0.5^ relation.

This approximate proportionality is observed over a wide range of scales, from the largest universe scales of galaxy clusters, to galaxy scales, and down to the much smaller scales of individual stars. In galaxy clusters, most of the baryons (60–90%) are found in the X-ray-emitting intracluster medium (ICM) at temperatures of 1–15 keV (approximately 10^7^ − 1.5 × 10^8^ K), with all the remaining baryons located in galaxies. Galaxy cluster masses over the two-decade range of 10^13^–10^15^ solar masses, M_⊙_, correspond closely with only a single decade of an ICM X-ray luminous temperature range of 1–10 keV, corresponding with *SDM*^0.5^ (see Figure 15 of [55]). ICM accounts for most of the baryon entropy in the universe [56], and may make a significant contribution to IDE. The halo mass–temperature relation shows a similar temperature for α SDM^0.6^ over a range of over three decades of mass for galaxy clusters, groups of galaxies, and individual galaxies (see Figure 4 of [57]; in that publication, it is deduced as mass proportional to *T*^1.65^). At the other extreme of universe scale (in the main sequence stars), the temperature is again α SDM^0.5^ because, over the two decades of star size from 0.5 M_⊙_ to 60 M_⊙_, the stellar photosphere temperature ranges over just one decade from 3800 to 44,500 K.

The two CPL *w*_0_ − *w*_a_ parameter combinations that best describe SMD(a)^+1.0^ and SMD(*a*)^+0.5^ show, respectively, a reasonable and a very good overlap with the *w*_0_ − *w*_a_ limits placed by dark energy measurements. This strongly supports the suggestion that there is a dynamic, phantom dark energy that is directly related to star/structure formation and primarily determined by temperature. Clearly, IDE can explain such a relation.

So far, we have shown that IDE(a) varies over time, with a relative variation that is consistent with the dark energy experimental results in Figure 5 and Figure 6. We now need to show that the absolute energy density expected from IDE is sufficient to explain the observed dark energy effects. In Figure 7, we provide a survey of possible sources of information energy [16,17,56], with total information energies of these phenomena determined by their *NT* product. Their information energy relevance is illustrated by a comparison with the ~10^70^ joules mc^2^ energy equivalence of the 10^53^ kg universe baryons. While there is much uncertainty in these *NT* values, it appears that the strongest sources of IDE—provided by stellar heated gas and by the intracluster medium—are approaching the baryon mc^2^ energy. Therefore, this work establishes that IDE can account for the dark energy of the universe, very approximately accounting for the present energy density, but, more clearly and importantly, providing a clear agreement with the latest experimentally observed dark energy history in *w*_0_ − *w*_a_ space [22,23,24].

### 4.2. Cause and Effect

We have established a clear similarity between the history of dark energy and the history of star formation via the *w*_0_ − *w*_a_ parameter plots. This has led us to consider IDE as the source of dark energy. However, this similarity could also be explained by the reverse possibility, that dark energy might be responsible for some of the SMD(*a*) history. Indeed, the very recent reduction in star formation, reduction in galaxy merging, and reduction in general structure growth rate have been attributed to the accelerating expansion caused by dark energy [58,59]. This reduction in SMD(*a*) is evident in Figure 2 by the decrease down to *z* = 0 shown by the fitted CPL red curves, corresponding with a present equation of state parameter of *w*_0_ > −1. However, we should expect that this would also happen if IDE is the source of dark energy. Increasing the star formation provided an increase in IDE energy density that eventually overtook the falling matter energy density to initiate the accelerating expansion. In turn, acceleration fed back to limit the star formation rate and IDE.

Over the late-universe history considered here, *a* > 0.2, there are two factors that support the main causal direction of SMD → IDE→ accelerating expansion. The first is the fact that SMD*(a)^+0.5^* provided the best fit in all the *w*_0_ − *w*_a_ parameter plots, corresponding with the temperature dependence expected for IDE rather than a mass dependence. The second is the close similarity in earlier growth rates, or similar *w*_a_ values, before dark energy was strong enough (relative to falling mass density) to affect the general structure growth rate.

### 4.3. IDE Should Resolve Some Problems and Tensions of ΛCDM

Late-universe dark energy in the form of information dark energy was previously shown [15] to be able to address many problems and tensions of ΛCDM. Some of the most relevant are briefly restated here. A dark energy theory that can account for the observed effects of dark energy as an alternative to the cosmological constant would allow Λ to take the more likely zero value [60] and effectively solve the cosmological constant problem by Λ→0. Also, we see in Figure 6 that the cosmological constant (*w*_0_ = −1 and *w*_a_ = 0) generally lies outside, or in some cases, only on the margins of, the *w*_0_ − *w*_a_ likelihood space of the recent experiments [22,23,24].

Late-universe dynamic dark energy that increases from *z ~* 2 to the present energy density has been previously shown [61] to provide a possible explanation for both the Hubble and σ_8_ tensions, similar to a transitional dark energy [62] or to a dynamic cosmological constant/running vacuum model [63]. The relatively fast increase in star formation between z = 2 and today combined with the *a^−3^* fall in matter energy density effectively provides suddenly significant dark energy as it was previously insignificant at z > 2 relative to the matter energy density. The time history of IDE described here would have a very similar characteristic time history to a sudden turn-on of dark energy [64] and thus may also account for both tensions.

Dark energy that increased with star formation to become the dominant energy today naturally solves the cosmological coincidence problem. Increasing star formation also increased the probability of intelligent beings evolving to live in the dark-energy-dominated epoch, and to subsequently discover dark energy. Note that the observed value of Λ has been shown [65] to be too small to be compatible with a comparable anthropic reasoning when assuming ΛCDM.

### 4.4. IDE Can Also Account for Dark-Matter Attributed Effects

At the time of writing, the latest results from the most sensitive WIMP dark matter detector to date [66] managed to further limit the possible WIMP dark matter energy range, and still without any confirmed dark matter particle detection. Now, IDE has similar universe-wide total energy as matter and is primarily concentrated around stars and structures where it should have a local energy density at least as high as that from baryons. The General Theory of Relativity shows that space–time will be distorted by accumulations of energy in any form, not just by the *m*c^2^ of matter, as illustrated in Figure 8.

Therefore, IDE in galaxies will have the same effect as an extra unseen dark matter component and will thus be difficult to distinguish from dark matter. The dark matter attributed effects in many galaxies have been shown to have their location fully specified by the location of baryons [67,68]. This observation is difficult to reconcile with ΛCDM, but it is clearly compatible with the effects expected from IDE and is also compatible with modified Newtonian dynamics (MOND). In a cluster of galaxies, the brightest and highest temperature galaxy is often found to have the strongest dark matter attributed effects [69], again consistent with an IDE source of the effects. When galaxies collide, the dark matter attributed effects generally pass straight through, remaining co-located with the stars and structures, while the gas clouds slow down with collisions [70,71,72]. This is also consistent with an IDE source of the effects.

New observations of early emerging massive galaxies and early clusters of galaxies require an accelerating structure formation at those times. This has been shown [73] to be difficult for the linear hierarchical galaxy formation of ΛCDM and more in keeping with the non-linear effects predicted by MOND. Equally, we should also expect similar non-linear effects with an increase in local attraction provided by IDE increasing with star formation.

The ΛCDM model assumes the acceleration continues to increase in our dark energy dominated epoch towards an eventual universe heat death, but IDE will lead to a different future. The stellar mass density is clearly starting to stop increasing (Figure 1 and Figure 2). The future maximum star formation density may be only 5% above today’s density [74]. This is also compatible with an analysis [75] of DESI data [24], which showed that dark energy density “reaches its maximum value it will ever achieve within the observed window”. We can expect the IDE energy density to fall and, some time in the future, revert to a matter dominated epoch with deceleration. Perhaps this will eventually lead to a ‘big crunch’ or even an ‘oscillating’ or ‘bouncing’ universe.

IDE is the equivalent energy of the information carried by, or represented by, the matter in the structures of the universe. This close association of IDE with structures is not compatible with the normally assumed strong damping of dark energy perturbations. However, the IDE model leads us to expect the measured H_0_ to vary depending on the distribution of mass and temperature along the line of measurement, and there is some evidence for possible directional anisotropies in H_0_ [76] that would be expected with IDE. Also, distant quasars gravitationally lensed by closer galaxies yield H_0_ values dependent on the lens redshift. This and other observations indicate a need to consider the possibility of both new late-time and new local physics [77].

Overall, we expect IDE to also account for at least some of the effects previously attributed to dark matter. Then, locally, IDE is attractive like invisible dark matter, but, universe-wide, IDE is repulsive as dark energy. These results are summarised in a simplified form in Table 1, where IDE is compared with other sources of dark energy and dark matter.

## 5. Summary

When compared in *w*_0_ − *w*_a_ space, the measured stellar mass density history, SMD(*a*), makes a very good fit with all measurements of dark energy history. This suggests that dark energy is directly driven by star and structure formations. The preference of fit for SMD(*a*)^+0.5^ over SMD(*a*)^+1.0^ further suggests that dark energy density is primarily determined by the temperature of structures at many levels of scale, whether galaxy clusters, galaxies, or stars. The dark energy dependence on structure formation, in particular with the dependence on temperature, is compatible with an information dark energy (IDE) explanation.

## Figures and Tables

**Figure 1 entropy-27-00110-f001:**
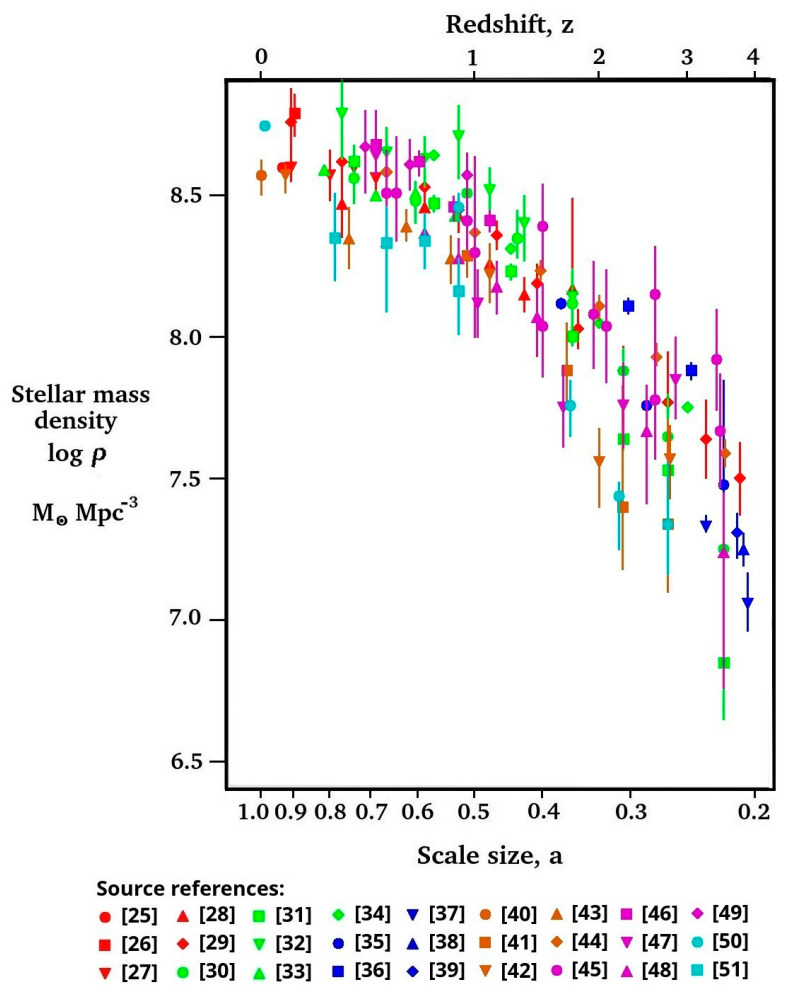
Survey of 121 measurements of stellar mass density measurements. Data sources: [25,26,27,28,29,30,31,32,33,34,35,36,37,38,39,40,41,42,43,44,45,46,47,48,49,50,51].

**Figure 2 entropy-27-00110-f002:**
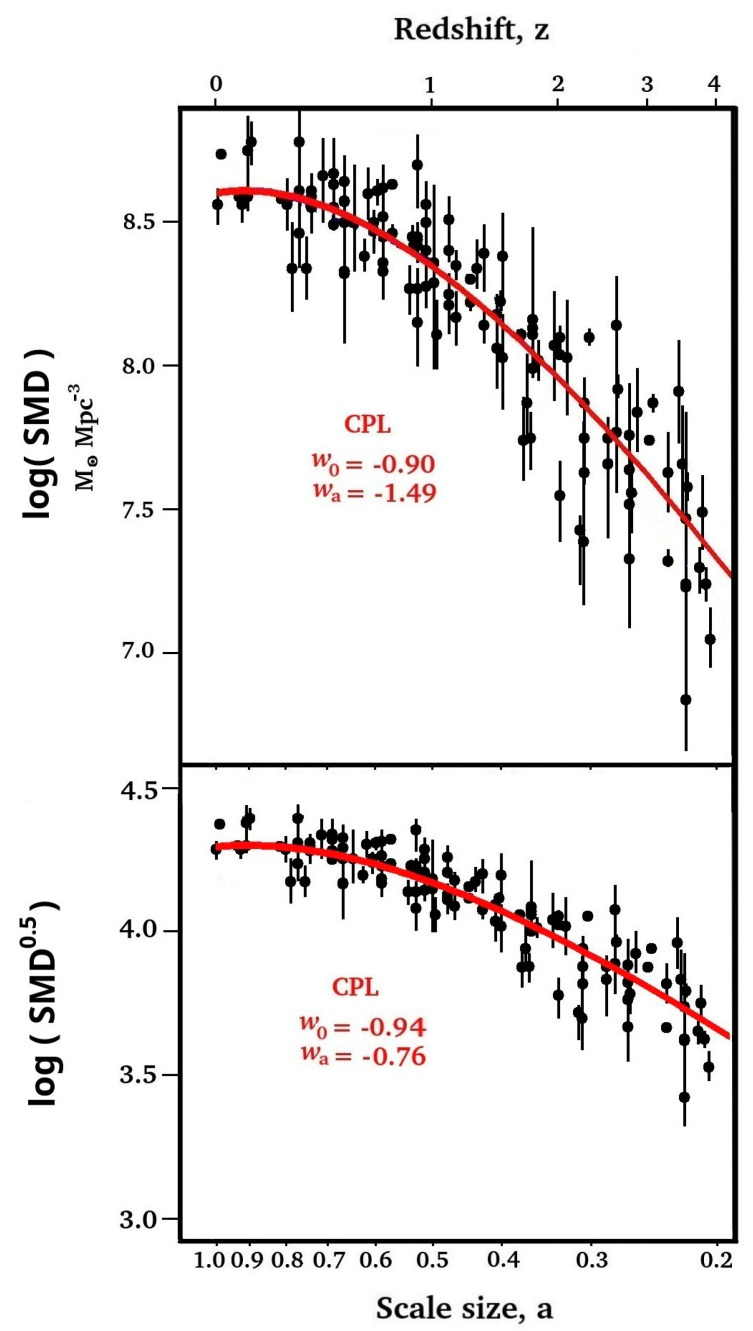
Measured stellar mass densities [25,26,27,28,29,30,31,32,33,34,35,36,37,38,39,40,41,42,43,44,45,46,47,48,49,50,51] plotted against universe scale size. The red curves correspond with the CPL best-fit parameters of *w*_0_ = −0.90 and *w*_a_= −1.49 for IDE(*a*) α SMD(*a*) and *w*_0_= −0.94 and *w*_a_= −0.76 for IDE(*a*) α SMD(*a*)^0.5^.

**Figure 3 entropy-27-00110-f003:**
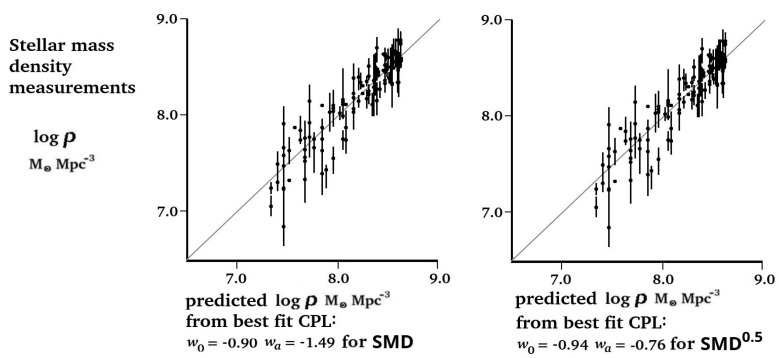
Identical plots of SMD measurements predicted by the best-fit CPL parameters against measured stellar mass densities at the same scale size for SMD and SMD^0.5^. Both plots show a Pearson correlation of r = 0.92.

**Figure 4 entropy-27-00110-f004:**
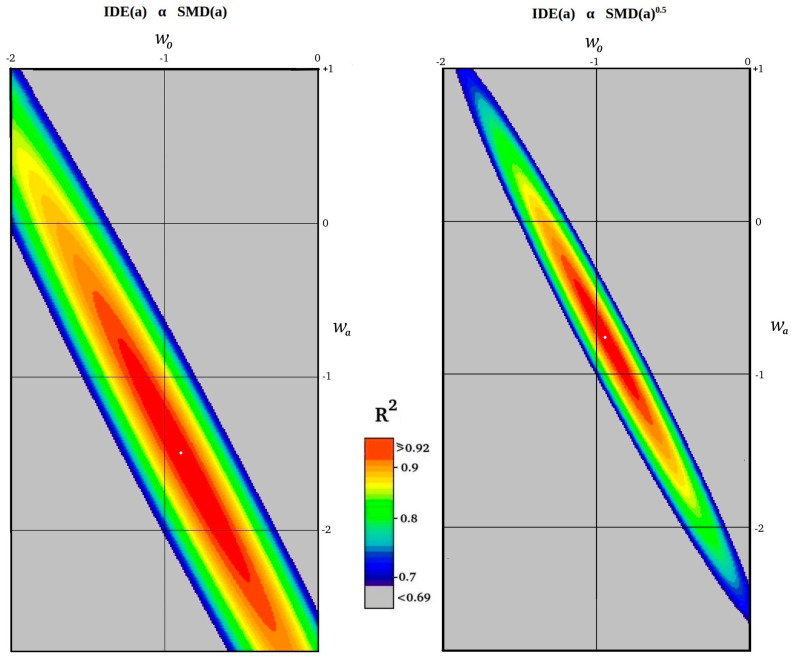
Plot of maximum R^2^ coefficient of determination values in *w*_0_ − *w*_a_ parameter space in parameter steps of 0.01 for SMD(*a*) and SMD(*a*)^0.5^. The best-fit *w*_0_ − *w*_a_ locations are identified by white dots.

**Figure 5 entropy-27-00110-f005:**
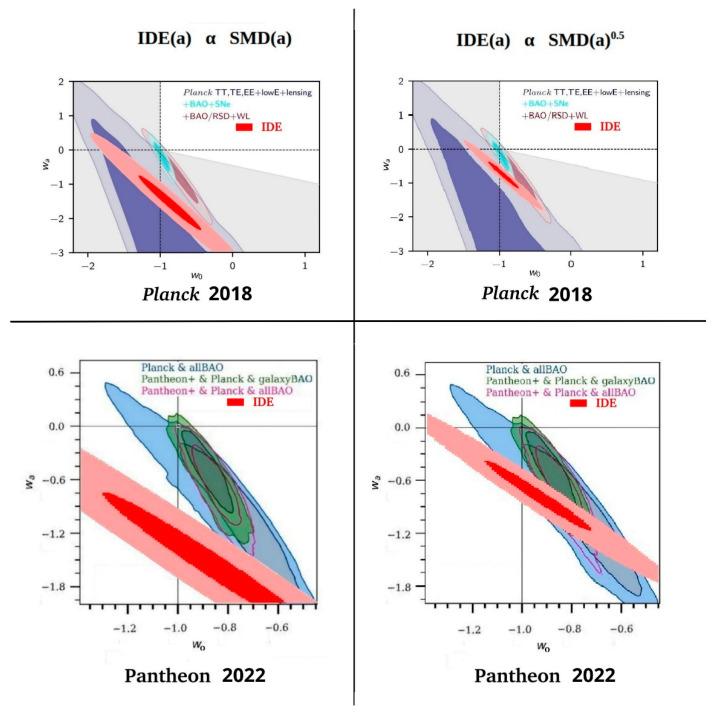
Information dark energy predicted contribution for the cases of IDE(*a*) α SMD(*a*) and IDE(*a*) α SMD(*a*)^0.5^ compared in *w*_0_ − *w*_a_ space with *Planck* 2018 and Pantheon+ 2022 results. Adapted from *Planck*, Figure 30 of [54], and Pantheon+, Figure 12 of [22], under the Creative Commons BY 4.0 License.

**Figure 6 entropy-27-00110-f006:**
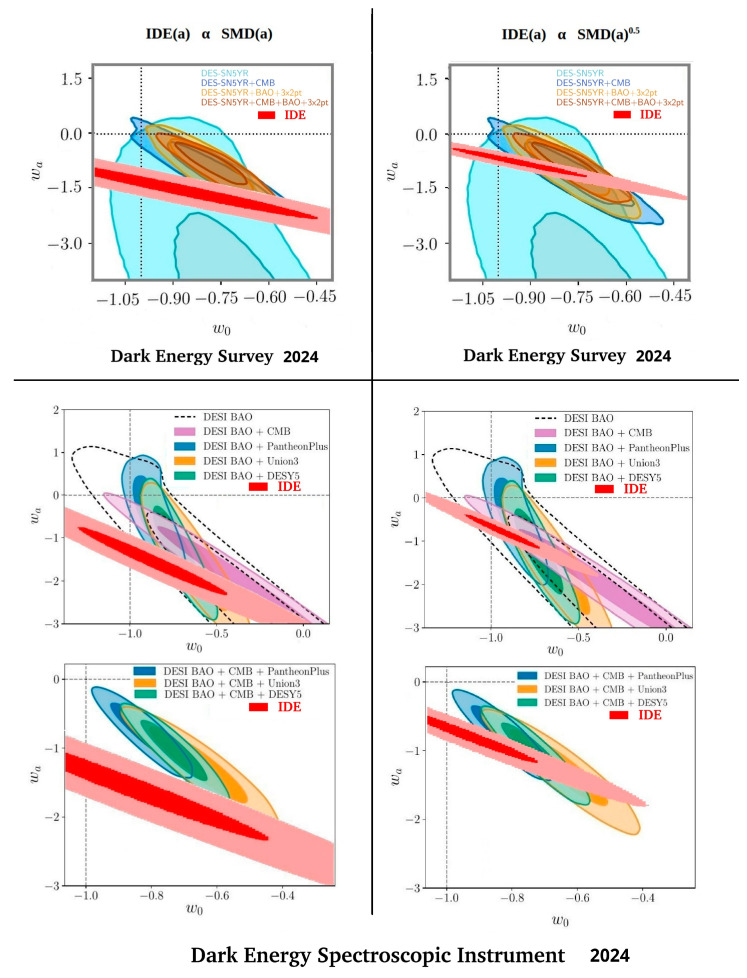
Information dark energy predicted contribution for the cases of IDE(*a*) α SMD(*a*) and IDE(*a*) α SMD(*a*)^0.5^ compared in *w*_0_ − *w*_a_ space with Dark Energy Survey 2024 and Dark Energy Spectrographic Instrument 2024 results. Adapted from Dark Energy Survey, Figure 8 of [23], and Dark Energy Spectroscopic Instrument, Figure 6 of [24], under the Creative Commons BY 4.0 License.

**Figure 7 entropy-27-00110-f007:**
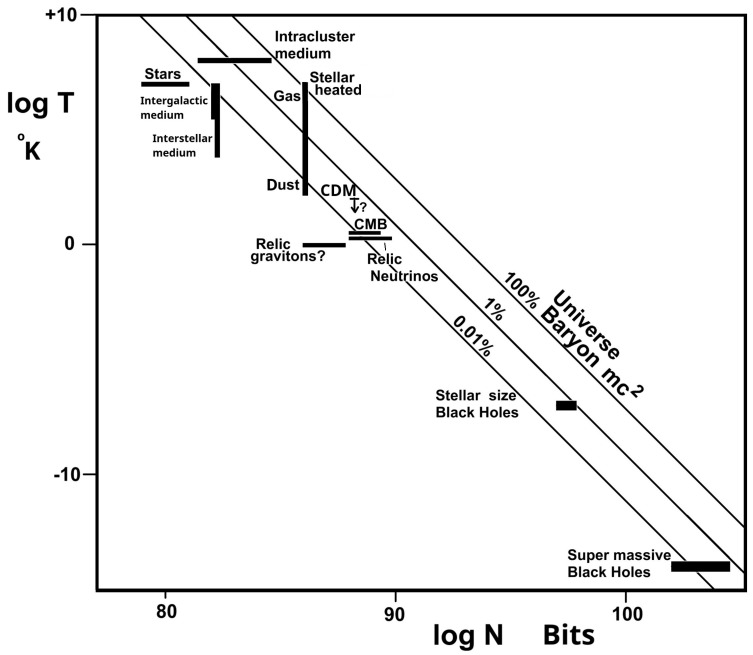
Sources of information energy with *NT* products compared with values that provide an information energy equivalent to the universe total baryon mc^2^.

**Figure 8 entropy-27-00110-f008:**
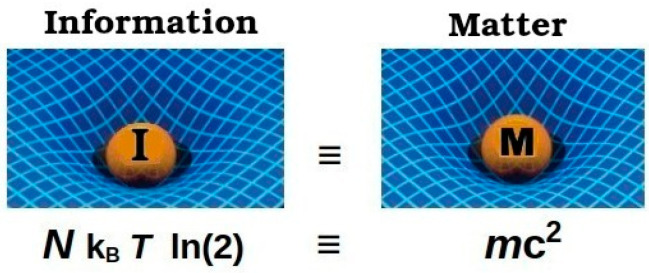
Space–time is equally distorted by an accumulation of energy in any form, comparing here the *N* k_B_ *T* ln(2) energy of information with an equivalent effect of *m*c^2^ energy of matter.

**Table 1 entropy-27-00110-t001:** Simple comparison of IDE with ΛCDM, scalar fields/quintessence, and MOND.

Required Dark Side Property	IDE	ΛCDM	Scalar Fields/ Quintessence	MOND
**Account for present** **dark energy density**	YES, order of **magnitude 10^70^ J**	NO, not by very many **orders of magnitude**	Only by much **fine tuning**	**-----**
**Consistent with** ** *w* _0_ * − w_a_* ** **experiment data**	YES, very good **agreement**	NO, on margins of **likelihood region**	Not specific **−1 < *w* < +1**	**----**
**Resolve Cosmological** **Constant problem**	YES, **Λ → 0**	**NO**	**Only by much fine tuning**	**----**
**Resolve Cosmological** **Coincidence problem**	YES, **naturally**	**NO**	Only by much **fine tuning**	**----**
**Resolve H_0_ and** **σ_8_** **Tensions**	YES, **possibly**	**NO**	**NO**	**----**
**Account for size of** **dark matter attributed** **effects**	YES, **same order of magnitude**	NO, dark matter not **even detected yet**	**----**	YES, **sometimes**
**Account for location of dark matter attributed** **effects**	YES coincident with **baryons**	NO, not coincident **with baryons**	**----**	YES, Coincident **with baryons**
**Account for early** **massive galaxies** **non-linear growth**	YES, **expect non-linear growth**	NO, only linear, **hierarchical growth**	**----**	YES, expect non-linear **growth**

## Data Availability

Data available in a publicly accessible repository.

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
