# Peer review of "Evidence for Dark Energy Driven by Star Formation: Information Dark Energy?"

_entropy, 2025, doi:10.3390/e27020110_

Round 1
Reviewer 1 Report
Comments and Suggestions for Authors
The author has for some time been proposing a possible connection between information and dark energy, particularly invoking the information erasure by star formation. While this remains a speculative avenue, it has received new impetus as it is one of the very few models that predicted in advance the trend of an increasing dark energy density (so called phantom dark energy) claimed as strongly hinted at in a data compilation analysed by the DEI collaboration. As such I am supportive of this article being published. However I feel ether are quite a few points that need clarification and that there can also be some improvements to the presentation, which I would like to see addressed before I could recommend it.
Scientific points:
1) A weakness is the lack of clarity as to how information originally stored in the very inhomogeneous stellar distribution can be considered as globally homogeneous like a dark energy component. In particular is this energy density to be held in some material substrate or not? The normal evolution (strong damping) of dark energy perturbations is due to an effective sound speed of the speed of light, or close to it, but that is not obviously the case here. Indeed the later discussion attempting to ascribe dark matter properties to the IDE seems to undermine this as that requires the energy density associated to the information to have essentially zero sound speed (so as to remain concentrated at its point of origin). There is a sense there that the author is trying to have things both ways, but the phenomenological properties of perturbation evolution of dark matter and energy are radically different and the data fits such as the CMB that the author shows require both. I would like to see a clearer discussions on these points.
2) When the effective CPL parameters are extracted in Figure 2, is Omega_IDE (equivalently Omega_M) varied in the fits? Why are we not shown the best-fit values?
3) While the CPL parametrisation is a useful point of reference, the proposed model feels more similar to the model of a sudden turn-on of Lambda. These sorts of scenarios have been discussed for some time; sources can be located via the recent paper arXiv:2406.07526 for example. Can a connection be made?
4) The CPL parametrisation is best interpreted by using the pivot scale at which the normalisation and slope decorrelate, which is where the data are strongest, see e.g. arXiv:2404.08056 for the case of DESI. This removes the strong correlation between the amplitude and slope and makes clear that all the evolution hint is in w_a and not the amplitude. However the effective CPL contours of the model seen in e.g. Fig 7 do not appear to follow the usual pivot degeneracy, which is interesting. Can these considerations shed more light?
5) Is there any relation of the current work to recent applications of Landauer to the cosmic horizon, eg arXiv:2407.15231 and arXiv:2409.05009?
Style points:
6) The constant k in the abstract has not been given a definition, which is needed at least in words for the asbtract to be self-contained.
7) The Fig 2 graphs are very ugly with mismatched aspect ratios. Since log(SMD^0.5} is just 0.5 log(SMD), I anyway do not see any reason not to put both lines on the same graph?
8) There is no reason to include all three Planck epochs, since 2018 is not only is the definitive data set but makes improvements to the analysis compared to earlier releases that contained less data and used older external datasets. Only the comparison with this final data is useful and should be shown.
Author Response
Reviewer 1 Comments & Responses
1) A weakness is the lack of clarity as to how information originally stored in the very inhomogeneous stellar distribution can be considered as globally homogeneous like a dark energy component. In particular is this energy density to be held in some material substrate or not? The normal evolution (strong damping) of dark energy perturbations is due to an effective sound speed of the speed of light, or close to it, but that is not obviously the case here. Indeed the later discussion attempting to ascribe dark matter properties to the IDE seems to undermine this as that requires the energy density associated to the information to have essentially zero sound speed (so as to remain concentrated at its point of origin). There is a sense there that the author is trying to have things both ways, but the phenomenological properties of perturbation evolution of dark matter and energy are radically different and the data fits such as the CMB that the author shows require both. I would like to see a clearer discussions on these points.
Response
I have noted this problem in the text lines 370-377. I realise the significance of the problem and do not wish to underestimate it, but have concentrated in this work on showing the compatibility of IDE to observations. I have added that there is some evidence for different H0 in different directions ref 76, to be expected with an IDE locked to structures, and included ref 77 which argues for both new late-time and new local physics to resolve the H0 tension.
2) When the effective CPL parameters are extracted in Figure 2, is Omega_IDE (equivalently Omega_M) varied in the fits? Why are we not shown the best-fit values?
Response
I have tried to clarify the fitting in the text lines 159-167. The best fit CPL values describe the history of IDE relative to today because of the division of SMD(a) by SMD(1.0). The Ω terms, today’s fractions of the total energy density, are unaffected. Figure 2 red curves are the overall best fit w0-wa value combinations in the wide range considered in figure 4. The best fit values are now also indicated on Figure 4 with small white dots.
3) While the CPL parametrisation is a useful point of reference, the proposed model feels more similar to the model of a sudden turn-on of Lambda. These sorts of scenarios have been discussed for some time; sources can be located via the recent paper arXiv:2406.07526 for example. Can a connection be made?
Response
In lines 315-322, I have now included a reference to the sudden turn-on of Λ model. It is similar to the transitional dark energy and running vacuum models previously mentioned and is now referenced along with those in the discussion of section 4.3. Note the expected time variation of IDE is governed by SMD(a) and the data fits of figure 2 shows that CPL does provide a good description of this expected time variation. The relatively fast increase in star formation between z=2 and today, combined with the a-3 fall in matter energy density, provides effectively a suddenly significant IDE whilst previously insignificant at z=2, relative to the matter energy density.
4) The CPL parametrisation is best interpreted by using the pivot scale at which the normalisation and slope decorrelate, which is where the data are strongest, see e.g. arXiv:2404.08056 for the case of DESI. This removes the strong correlation between the amplitude and slope and makes clear that all the evolution hint is in w_a and not the amplitude. However the effective CPL contours of the model seen in e.g. Fig 7 do not appear to follow the usual pivot degeneracy, which is interesting. Can these considerations shed more light?
Response
I have not felt comfortable applying the suggested pivot scale. However, I have now included a reference to the conclusion of arXiv:2404.08056. In my previous paper (ref 15) I had noted that star formation and hence IDE is reaching a peak and how this would affect the future universe. In lines 361-368 I have now referenced the work of arXiv:2404.08056 using a pivot scale as it effectively shows that DE reaches its “maximum value it will ever achieve within the observed window”. I have indicated that later this will lead to a different future than the heat death predicted by É…CDM. It will lead to a return to a matter dominated epoch with deceleration and eventually either a “big crunch” or bouncing/oscillating universe.
5) Is there any relation of the current work to recent applications of Landauer to the cosmic horizon, eg arXiv:2407.15231 and arXiv:2409.05009?
Response
I have noted the applications of the Landauer Principle to the cosmic horizon, but prefer to restrict the paper to considering the in-situ equivalent energy. I have included a new clearer explanation to alongside equation 2.
Style points:
6) The constant k in the abstract has not been given a definition, which is needed at least in words for the asbtract to be self-contained.
Response
In order to avoid confusion with curvature, changed the power, k, of SMD to p for equation 2 and elsewhere made it clear, for the two cases considered, by using just SMD and SMD0.5 in the abstract and all text and diagrams.
7) The Fig 2 graphs are very ugly with mismatched aspect ratios. Since log(SMD^0.5} is just 0.5 log(SMD), I anyway do not see any reason not to put both lines on the same graph?
Response
I have changed Fig 2 appropriately.
8) There is no reason to include all three Planck epochs, since 2018 is not only is the definitive data set but makes improvements to the analysis compared to earlier releases that contained less data and used older external datasets. Only the comparison with this final data is useful and should be shown.
Response
Agreed- I have removed the two earlier Planck plots and their references. The old figures 5,6,7 are now just figure 5 (2018 & 2022 data) & Fig 6 (latest 2024 data: DES & DESI). Also added the required Creative Commons License to the figures.

Reviewer 2 Report
Comments and Suggestions for Authors
I have the following points that should be considered prior to give my recommendation for publication:
1. The mathematical model is not clearly explained. On the one hand, eq. (2) introduces the IDE dynamics, but apart from this equation there is no connection to the SMD density. This latter is not specified: What is mathematical form of SMD? The red curve in fig. 2. Is SMD varying as 1/a^3?. In fact, formally this model is an interacting model of IDE and SMD that should comply with the continuity equation and should affect the SMD evolution due to IDE.
2. Explain how the IDE model as introduced in eq. (2) is related to the CPL parametrization.
3. In figure 2, what is the methodology used to plot the data for different curvatures? That is why data points are different for different curvatures?
4. Clarify the method used to fit the data (R^2 parameter). It looks that you did not use a standard MCMC method to vary the parameters and to minimize a chi^2 function.
5. It is known that the curvature is close to flat in LCDM model, but you have chosen to nonzero values of k, why? Apart, do not use k, but instead Omega_k, as this latter is a quantitative parameter that can be compared with other results, e.g. Planck’s.
6. Plots 5-7 are not professional, it looks like your results were simply put over other collaboration results. One should run the chains of other collaboration results and run your own chains and present all them together in a consistent way, not simply gluing your contour plots to other results.
7. Figure 8 has no quantitative relationship with the previous results, so I do not see the need to put it there. The arguments from figure 8 should perhaps be mentioned in the introduction.
8. I do not see the point to include sections 4.2-4.4, that are not based on computations made in the present work.
9. The second paragraph of the summary is not based on the present results, they sound more as speculations. The summary should summarize the main results of the computations.
Author Response
Reviewer 2 Comments and Responses
1. The mathematical model is not clearly explained. On the one hand, eq. (2) introduces the IDE dynamics, but apart from this equation there is no connection to the SMD density. This latter is not specified: What is mathematical form of SMD? The red curve in fig. 2. Is SMD varying as 1/a^3?. In fact, formally this model is an interacting model of IDE and SMD that should comply with the continuity equation and should affect the SMD evolution due to IDE.
Response
Combined answer below to points 1 & 2.
2. Explain how the IDE model as introduced in eq. (2) is related to the CPL parametrization.
Response
The energy densities of all matter and dark energy in equations 1 & 2 are normalised to their present relative energy densities given by the Ω terms (Ω terms add up to 1). While the total (dark + baryon) matter energy energy densities vary as a-3 ( so Ωtot a-3 in equations 1 &2 ), we expect É… to be constant (so just ΩÉ… in equation 1) but in figure 2 we expect IDE will vary with some power of SMD(a). We choose to study SMD+1 and SMD+0.5. The stellar mass density, SMD(a), is just a series of measurements (Fig 1) at different universe scale sizes, a. For compatibility with experimental measurements of dark energy we must find the best CPL fits to describe the variation of SMD+1 and SMD+0.5. SMD the stellar mass density, which is proportional to the fraction of baryons which have formed stars, has increased for most of the universe timeline. CPL is just a common tool used in cosmology to compare dark energy relative variation with time (universe scale size, a where a=1 today).
CPL parameters describe how the equation of state parameter w varies with scale size, a : w(a)=w0 +(1-a) wa
and, in turn, the equation of state parameter for a dark energy determines how the dark energy density varies as a function of scale size, a : as a-3(1+w)
This is defined in lines 72+
3. In figure 2, what is the methodology used to plot the data for different curvatures? That is why data points are different for different curvatures?
Response
Note my response to your point 5 below. In figure 2 it is not different curvatures but different powers of SMD(a). My original choice of k for power was unfortunate. As information energy density is dependent on temperature and as it depends on amount of information in a given volume, we expect it to vary in some proportionality to the stellar mass density, SMD. We choose to investigate two cases IDE α SMD and IDE α SMD0.5 . The case for SMD0.5 is based on the observation that the temperature of various astrophysical objects is proportional to their mass0.5.
4. Clarify the method used to fit the data (R^2 parameter). It looks that you did not use a standard MCMC method to vary the parameters and to minimize a chi^2 function.
Response
For comparison with experimental data presented in terms of CPL parameters it is necessary to also use the CPL parameterisation. As the CPL parameterisation limits the set of defined curves, I have found the best fit by simply minimising the residual sum of squares, RSS=Σi (SMD(ai) – f(ai))2
where SMD(ai) is the ith measurement of stellar mass density, ai is the scale size at that measurement, and f(ai) the CPL predicted value of SMD at that scale size.
This is now made clearer in the text lines 159-167 and figures 2 & 4.
5. It is known that the curvature is close to flat in LCDM model, but you have chosen to nonzero values of k, why? Apart, do not use k, but instead Omega_k, as this latter is a quantitative parameter that can be compared with other results, e.g. Planck’s.
Response
Again I must apologise for the confusion caused by my original choice of k for the power of SMD. The parameter k in the original paper was not the cosmological curvature term but the power of SMD(a). In order to remove this confusion I have changed the immediate text around eq 2 to use p for the power. Elsewhere in the text and diagrams where we are just using two values for p (1 and 0.5) I have used simply SMD and SMD0.5 for maximum clarity.
6. Plots 5-7 are not professional, it looks like your results were simply put over other collaboration results. One should run the chains of other collaboration results and run your own chains and present all them together in a consistent way, not simply gluing your contour plots to other results.
Response
This presentation comparison is the standard way experimental data and theoretical models are considered in cosmology and astrophysics (e.g. see Fig 1 of reference 75). Then the w0-wa experiment and theory plots relative location and overlap are clear. The old figures 5-7 are now replaced by new figures 5,6 that remove the two of the early Planck plots and no longer use the parameter k.
7. Figure 8 has no quantitative relationship with the previous results, so I do not see the need to put it there. The arguments from figure 8 should perhaps be mentioned in the introduction.
Response
I have now made the requirement for this clearer in the associated text lines 265+.
Up to this point in the paper the emphasis has been on showing that the IDE model has the appropriate relative variation in time to explain the data. Old Figure 8 ( new Fig. 7 ) is essential to also show that estimated sources of IDE are close to providing NT product values that can account for the required absolute energy densities.
8. I do not see the point to include sections 4.2-4.4, that are not based on computations made in the present work.
Response
It is essential and standard practice to consider what the effects of such a model would be: whether those effects are consistent with other types of observation; make a prediction for future measurement, or have the potential to solve an existing problem/tension.
9. The second paragraph of the summary is not based on the present results, they sound more as speculations. The summary should summarize the main results of the computations.
Response
Agreed - I have now reduced the summary to just emphasize the main result.

Round 2
Reviewer 1 Report
Comments and Suggestions for Authors
Thank you for the updates which I find have made teh article significantly clearer and ready for publication.
Reviewer 2 Report
Comments and Suggestions for Authors
The author addressed my points, so I can recommend now the manuscript for publication.